ⓐ | **Open Peer Review** | Bacteriology | Research Article

# Involvement of an orphan response regulator of the two-component regulatory system in the formation of physiologically mature sporangia in *Actinoplanes missouriensis*

Takuya Akutsu,[1] Zhuwen Tan,[1] Aiko Hirata,[2] Takeaki Tezuka,[1,3] Yasuo Ohnishi[1,3]

**ABSTRACT**   The actinomycete *Actinoplanes missouriensis* forms terminal sporangia that contain dormant sporangiospores. Upon contact with water, sporangia release zoospores through a process called sporangium dehiscence. In this study, we characterized *asfR* (*AMIS_76070*), which encodes an orphan response regulator receiver domain protein of the two-component regulatory system, as one of 136 genes whose transcription was highly activated during sporangium formation. A̲ctinoplanes s̲porangium f̲ormation r̲egulator (AsfR) homologs are conserved among *Actinoplanes* bacteria. An *asfR* null mutant (Δ*asfR*) strain formed normally shaped sporangia containing apparently normal dormant spores, but they exhibited defective sporangium dehiscence; the number of spores released from the sporangia of the Δ*asfR* strain was four orders of magnitude lower than that from the sporangia of the wild-type strain. This phenotypic change was recovered by introducing *asfR* with its own promoter into the Δ*asfR* strain. Based on the amino acid sequence and predicted structure, the function of AsfR appeared to be controlled by the phosphorylation of Asp-72. Consistently, the phenotypic change observed in the Δ*asfR* strain was not restored by introducing a mutated *asfR* (D72N) gene. Three orphan histidine kinases (HKs) in *A. missouriensis* were found to interact with AsfR by screening using a bacterial two-hybrid assay. However, gene disruption experiments revealed that these three HKs were not required for sporangium dehiscence in *A. missouriensis*. Although the molecular functions of AsfR remain to be elucidated, this study shows that AsfR is involved in the formation of physiologically mature sporangia that are fully prepared to release spores under sporangium dehiscence-inducing conditions.

**IMPORTANCE** *Actinoplanes missouriensis* undergoes a life cycle involving complex morphological development, including mycelial growth, sporangium formation and dehiscence, swimming as zoospores, germination, and outgrowth to mycelial growth. In this study, we revealed that a stand-alone response regulator receiver domain protein, AsfR, is required for the formation of physiologically mature sporangia that can release spores under sporangium dehiscence-inducing conditions. *A. missouriensis* seems to express genes that are involved in sporangium dehiscence during sporangium formation, considering that an *asfR* null mutant produced normally shaped sporangia, but these sporangia were deficient in sporangium dehiscence. Although the molecular functions of AsfR, as well as the histidine kinase(s) that phosphorylates AsfR, remain to be elucidated, identification of AsfR as a possible key regulator for the preparation of the onset and progression of sporangium dehiscence is significant, because almost no proteins involved in the early stages of sporangium dehiscence have been identified in *A. missouriensis*.

Address correspondence to Takeaki Tezuka, atezuka@g.ecc.u-tokyo.ac.jp, or Yasuo Ohnishi, ayasuo@g.ecc.u-tokyo.ac.jp.

The authors declare no conflict of interest.

See the funding table on p. 10.

**KEYWORDS** *Actinoplanes missouriensis*, sporangium formation, orphan response regulator, two-component regulatory system

The two-component regulatory system (TCS) mediates signal transduction, which allows bacterial organisms to sense and respond to changes in environmental conditions. An ordinary system is composed of a sensor histidine kinase (HK), which detects input information, and a response regulator (RR) that controls the output (1, 2). RRs are defined by the presence of an RR receiver domain, but most contain one or more output (effector) domains in addition to the receiver domain. Generally, the primary function of the receiver domain is to act as a phosphorylation-dependent switch within RRs. Despite their conserved domain structure, RRs are versatile because the receiver domain can be paired with remarkably different output domains. Receiver domains that are not paired with output domains have also been reported. These stand-alone receiver domain proteins presumably exert their roles by binding to their target proteins or by carrying the phosphoryl group between HKs and histidine phosphotransfer (Hpt) proteins in multistep phosphorelays (3, 4). Meanwhile, sensor HKs that are attached to the RR receiver domain are grouped into "hybrid HKs" (5). Predicting the primary role of stand-alone receiver domain proteins, based on their amino acid sequences, is challenging.

*Actinoplanes missouriensis* is characterized by complex morphological development. Following vegetative growth by extending branched substrate mycelia, *A. missouriensis* forms a few hundred dormant spores inside each terminal sporangium that arises from the substrate mycelium through a short sporangiophore. Inside the sporangium, spores are encapsulated in an intrasporangial matrix called the sporangium matrix. Spores are released from the sporangia upon contact with water via a process known as sporangium dehiscence (6, 7). After release from sporangia, spores swim as zoospores at a high velocity using flagella. Zoospores exhibit chemotactic behavior toward various types of compounds, such as sugars, amino acids, aromatic compounds, and mineral ions. Although zoospores can keep swimming for approximately an hour, they stop swimming to germinate by the function of a protein named FtgA, which is encoded within one of three chemotaxis gene clusters (8), and grow into the substrate mycelium. Under laboratory conditions, *A. missouriensis* forms sporangia when cultivated on a humic acid-trace element (HAT) agar medium. On this agar medium, small sporangium-like structures are produced after 2 days or 3 days of cultivation at 30°C. Mature sporangia that can release spores under sporangium dehiscence-inducing conditions are formed after incubation for 5–7 days (9, 10). Owing to its complex life cycle, *A. missouriensis* is an ideal species for studying the molecular mechanisms of morphological development in prokaryotes. In particular, sporangium dehiscence can be considered as the initial stage of the transition from dormant spores to actively growing cells. Therefore, the formation of physiologically mature sporangia that can release spores under sporangium dehiscence-inducing conditions is of great interest.

Previously, we performed a transcriptome analysis by RNA sequencing (RNA-Seq) using total RNAs extracted from *A. missouriensis* wild-type cells cultivated on HAT agar for 1, 3, 6, and 40 days to obtain the transcriptional profiles of developmental genes (11). Under the culture conditions, *A. missouriensis* grew as substrate mycelia on day 1 and then began to form sporangia on day 3. The majority of sporangia fully matured on day 6, as determined by scanning electron microscopy (SEM). This analysis revealed that the transcript levels of 136 genes in mycelia (with premature sporangia) grown on HAT agar for 3 days were at least 100-fold higher than those in mycelia grown for 1 day. These included 29 genes of the flagellar gene cluster, 4 genes of the pilus gene cluster, 9 methyl-accepting chemotaxis protein (MCP)-encoding genes, 2 MCP-like protein-encoding genes, and 3 regulatory genes involved in sporangium formation and dehiscence (*tcrA*, *hhkA*, and *fliA2*) (12–14). Meanwhile, the involvement of the remaining 89 genes in morphological development remains to be elucidated. In this study, we focused on one of the 89 genes, *AMIS_76070*, which encodes an RR receiver domain protein. Hereafter,

we designate AMIS_76070 as AsfR (*A*ctinoplanes *s*porangium *f*ormation *r*egulator) for its function revealed in this study. Here, we describe the functional analysis of AsfR by gene disruption and search for its partner HK(s). The findings of this study demonstrate that AsfR plays an essential role in the formation of physiologically mature sporangia that can release spores under sporangium dehiscence-inducing conditions.

## MATERIALS AND METHODS

### General methods

The bacterial strains, plasmid vectors, and media used in this study have been described previously (15–17). The primers used in this study are listed in Table S1. *A. missouriensis* was cultivated as previously described (12). SEM was performed using an S-4800 electron microscope (Hitachi, Tokyo, Japan) as previously described (18). Transmission electron microscopy (TEM) was performed using an H-7600 electron microscope (Hitachi) as described previously (19). Phase-contrast microscopy was performed using a BH-2 phase-contrast microscope (Olympus, Tokyo, Japan) as described previously (13). Free zoospores were quantified as previously described (20).

### Construction of gene deletion mutants

To construct an *asfR* null mutant (Δ*asfR*) strain, the upstream and downstream regions (approximately 2 kbp each) of *asfR* were amplified using PCR. The amplified DNA fragments were digested with appropriate restriction enzymes (Table S1) and cloned into pUC19 digested with the same restriction enzymes. The generated plasmids were sequenced to confirm the absence of PCR-derived errors. The cloned fragments were digested with restriction enzymes and cloned together into pK19mobApr (pAT19) digested with restriction enzymes (11, 21). The generated plasmid was introduced into the *A. missouriensis* wild-type strain by conjugation, as described previously (14). Apramycin-resistant colonies resulting from single-crossover recombination were isolated. One of them was grown in peptone-yeast extract-MgCl$_2$ liquid broth at 30°C for 48 h, and the mycelia, suspended in 0.75% NaCl solution, were spread onto the Czapek-Dox broth agar medium (BD, NJ, USA) containing extra sucrose (final concentration 5%). *A. missouriensis* cells expressing *sacB*, encoding levansucrase, from pK19mobApr exhibit lethal sensitivity to sucrose. After incubation at 30°C for 5 days, sucrose-resistant colonies were inoculated on yeast extract-beef extract-NZ amine-maltose monohydrate (YBNM) agar medium with or without apramycin to isolate apramycin-sensitive colonies resulting from the second single-crossover recombination as candidates for the gene deletion strains. Disruption of *asfR* was confirmed by Southern blotting (Fig. S1), as described previously (16). Single deletion mutants of three putative HK genes (*AMIS_17390*, *AMIS_33880*, and *AMIS_36100*) were generated using the above-described method; however, pK19mob*sacB* (16) was used instead of pK19mobApr (pAT19). Similarly, double-gene and triple-gene deletion mutant strains were constructed using single-gene and double-gene deletion mutant strains as parental strains, respectively. Disruption of the target genes was confirmed by PCR (data not shown).

### Construction of strains for gene complementation testing

A 0.5 kbp DNA fragment containing the promoter and coding sequences of *asfR* was amplified by PCR. The amplified fragment was digested with EcoRI and XbaI and cloned into pUC19 digested with the same restriction enzymes. The generated plasmid was sequenced to confirm that no PCR-derived errors were present. The cloned fragment was digested with EcoRI and XbaI and cloned into pCAM2 (22) digested with the same restriction enzymes. To construct the mutated *asfR* gene, pCAM2 carrying *asfR* was used for site-directed mutagenesis. The D72N mutated DNA fragment was amplified by overlap extension PCR using the primer pairs 76070_comp-F and 76070_D72N-R, and 76070_D72N-F and 76070_comp-R (Table S1). The amplified DNA fragment was digested

with EcoRI and XbaI and then cloned into pUC19 digested with the same restriction enzymes. The generated plasmid was sequenced to confirm that no PCR-derived errors were present. The cloned fragment was digested with EcoRI and XbaI and then cloned into pCAM2 digested with the same restriction enzymes. The generated plasmids were introduced into the Δ*asfR* strain by conjugation as described previously (14). Plasmid pCAM2 was also introduced into the wild-type and Δ*asfR* strains to generate the vector control strains. Then, apramycin-resistant colonies were isolated.

## Bacterial two-hybrid assay

The bacterial two-hybrid assay was conducted using a BACTH system kit (Euromedex, Strasbourg, France) according to the manufacturer's instructions. To construct the T18 or T25 domain fusion plasmids, the coding sequences of *asfR* and 33 orphan HK genes were amplified by PCR. The DNA fragments were cloned into pUC19 and sequenced to confirm that no PCR-derived errors were introduced. Then, the cloned fragments were digested with restriction enzymes and cloned into the vectors pUT18 or pUT18C (*asfR*) and pKT25 or pKNT25 (HK genes). *Escherichia coli* BTH101 competent cells were co-transformed with the T18 and T25 domain fusion plasmids, and transformants were selected on Luria-Bertani (LB) agar medium (BD, NJ, USA) containing ampicillin and kanamycin. To test for protein-protein interaction, three individual colonies per assay were grown overnight in LB liquid broth with ampicillin and kanamycin. The cultures were spotted onto M63 agar medium containing maltose, isopropyl-β-D(-)-thiogalacto-pyranoside (IPTG), 5-bromo-4-chloro-3-indolyl-β-D-galactopyranoside (X-Gal), ampicillin, and kanamycin. The agar plates were incubated at 30°C for 4 days and photographed. To quantify protein-protein interaction, the transformants were grown overnight at 30°C in LB liquid broth with ampicillin and kanamycin. The cultures were inoculated into LB liquid broth with ampicillin, kanamycin, and IPTG and cultivated at 30°C for 40 h. β-Galactosidase activities were quantified as described previously (23).

## RESULTS

### AsfR is conserved among *Actinoplanes* bacteria

AsfR comprises 138 amino acids. A protein database search using InterPro version 101.0 (https://www.ebi.ac.uk/interpro/) showed that AsfR harbors a signal transduction RR receiver domain (IPR001789; residues 21–137). As described in the Introduction, most RRs of the TCS contain one or more output domains, the majority of which regulate transcription, in addition to the receiver domain. Contrary to these RRs, AsfR consists of a single receiver domain. Therefore, AsfR presumably exerts its function by binding to other protein(s) or by carrying phosphoryl groups between sensor HKs and Hpt proteins in multistep phosphorelays. Although TCS RRs are typically encoded near their cognate HK genes, they are also encoded by orphan genes. We consider AsfR an orphan RR because neither an HK nor an Hpt protein is encoded in the vicinity of *asfR*; genes encoding a putative subtilase-family protease and a protein with unknown function are located upstream (*AMIS_76080*) and downstream (*AMIS_76060*), respectively, of *asfR* (Fig. S1A).

We predicted the monomeric structure of full-length AsfR using the Alpha-Fold2 ColabFold version (https://colab.research.google.com/github/sokrypton/Colab-Fold/blob/main/AlphaFold2.ipynb) (24, 25). The prediction tool generated a three-dimensional model with a $(\beta\alpha)_5$ topology composed of five alternating β-sheets and α-helices (Fig. 1). In this model, AsfR folds into a central five-stranded parallel β-sheet (β1–β5) surrounded by two α-helices on one side (α1 and α5) and three on the other (α2–α4) (Fig. 1B). The overall structure is typical of RR receiver domains. In the receiver domain structures, there is a characteristic quintet of highly conserved residues (Fig. 1B). Three Asp residues, two of which (Asp-27 and Asp-28 for AsfR) follow β1 and the remaining one (Asp-72) is located at the end of β3, bind a divalent metal ion that is essential for the phosphotransfer reaction. The third Asp (Asp-72) is the phosphorylation

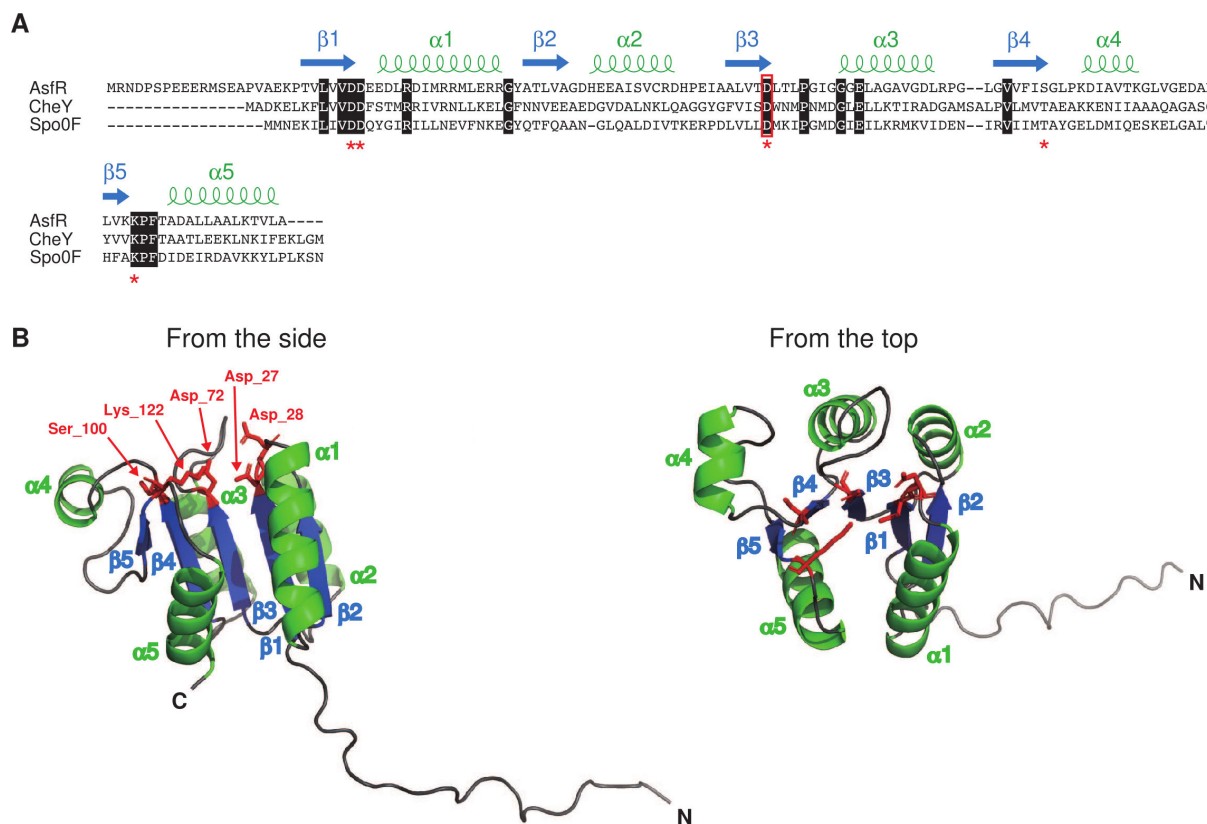

**FIG 1** Amino acid sequence alignment of AsfR, CheY, and Spo0F (A) and predicted structure of AsfR (B). (A) CheY and Spo0F are derived from *Escherichia coli* and *Bacillus subtilis*, respectively. The predicted secondary structures in the AsfR sequence are shown above the alignment. The characteristic quintet of the highly conserved residues is indicated by red asterisks. The Asp residue phosphorylated in CheY and Spo0F, which corresponds to Asp-72 in AsfR, is indicated by a red rectangle. Identical amino acid residues are shown in black background. (B) The polypeptide is shown by ribbon representation and colored green for α-helices (α1–α5) and blue for β-sheet regions (β1–β5). The remaining residues are colored gray. The five highly conserved residues in the receiver domain are shown in red (Asp-27, Asp-28, Asp-72, Ser-100, and Lys-122).

site. Typical RRs form homodimers using the α4-β5-α5 interface in response to the phosphorylation of this Asp residue. β4 ends in a highly conserved Ser (or Thr) residue (Ser-100), which is hydrogen-bonded to the phosphoryl group to induce phosphorylation-mediated conformational changes. A highly conserved Lys residue (Lys-122), located at the end of β5, is also important for phosphorylation-mediated conformational changes. These five residues are completely conserved in AsfR (Fig. 1). These results suggested that AsfR functions in a phosphorylation-dependent manner, and Asp-72 of AsfR is predicted to be phosphorylated by the phosphotransfer reaction.

We performed a BLAST search of the NCBI genome database (https://blast.ncbi.nlm.nih.gov/Blast.cgi) and found that AsfR homologs are conserved in at least 50 *Actinoplanes* species, including *A. missouriensis*, whose genome sequences and gene annotations have been registered in the database (Fig. S2A). Furthermore, proteins showing high similarities to AsfR are also conserved in other actinomycete species including *Patulibacter* sp. NPD 049589, *Pseudosporangium ferrugineum*, *Couchioplanes caeruleus*, *Nucisporomicrobium flavum*, *Spirilliplanes yamanashiensis*, *Krasilnikovia cinnamomea*, *Paractinoplanes lichenicola*, *Symbioplanes lichenis*, *Winogradskya consettensis* (Fig. S2B). Members of the genus *Patulibacter* form motile cells with long flagella (26). *C. caeruleus* forms flagellated arthrospores by fragmentation of aerial hyphae (27). *N. flavum* forms irregular pseudosporangia consisting of non-motile spores (28). *S. yamanashiensis* forms short chains of spores from aerial hyphae arranged in spirals (29). *K. cinnamomea* forms pseudosporangia on short sporangiophores above the surface

of substrate mycelium (30). Members of the genera *Paractinoplanes*, *Symbioplanes*, and *Winogradskya* are closely related to *Actinoplanes* (31). These homologs share at least 53% amino acid sequence identity with AsfR, demonstrating that AsfR homologs are evolutionarily conserved among the genus *Actinoplanes* and related actinomycetes (Fig. S2). Meanwhile, AsfR homologs are not conserved in members of the genus *Streptomyces* (less than 35% amino acid sequence identity), which forms branched substrate mycelia during vegetative growth and subsequently produces aerial hyphae that culminate in non-motile exospores (32).

## Transcription profile of *asfR* during *A. missouriensis* life cycle

As described in the Introduction, we focused on *asfR* because the transcript level of this gene was upregulated at the early stage of sporangium formation in our previous RNA-Seq analysis (11). Recently, we performed exhaustive RNA-Seq analysis at various time points during the life cycle of *A. missouriensis* (33). For this analysis, RNA samples for vegetative growth were prepared from substrate hyphae grown on YBNM agar for 1 day in triplicate. For sporangium formation, RNA samples were prepared from mycelia and/or sporangia grown on HAT agar for 1, 3, 6, and 15 days in triplicate at each time point. For sporangium dehiscence, we prepared RNA samples from sporangia (including some substrate hyphae) suspended and incubated in 25 mM histidine solution for 0, 15, and 60 min to induce sporangium dehiscence in triplicate at each time point. In this analysis, the transcription of *asfR* was highly upregulated during sporangium formation, whereas it was downregulated gradually during sporangium dehiscence (Fig. S3).

## AsfR plays a pivotal role in sporangium dehiscence

To examine the function of AsfR *in vivo*, we generated the Δ*asfR* strain. No difference was observed between the wild-type and Δ*asfR* strains by macroscopic observation of mycelia or sporangia formed on YBNM and HAT agar (data not shown). To examine sporangium formation in detail, we observed mycelia and sporangia of both strains grown on HAT agar at 30°C for 7 days using SEM. Both strains produced normal globose or subglobose sporangia with short sporangiophores (Fig. 2A and B). Previously, we reported that a null mutant strain of *hhkA* (Δ*hhkA* strain), which encodes a hybrid sensor HK involved in a signal transduction pathway for sporangium formation, spore dormancy, and sporangium dehiscence, produced many distorted spores of various sizes, including ectopically germinated spores inside the sporangium, as revealed by the observation of ultrathin sections of sporangia using TEM, whereas the Δ*hhkA* strain produced normal sporangia when observed by SEM (13). Therefore, we observed ultrathin sections of wild-type and Δ*asfR* sporangia grown under the same conditions as for SEM, using TEM to examine spore maturation inside a sporangium. Both strains produced normal round sporangiospores of similar sizes (Fig. 2C and D).

We examined sporangium dehiscence and motility of zoospores using phase-contrast microscopy. Under laboratory conditions, sporangium dehiscence can be induced either by pouring 25 mM $NH_4HCO_3$ solution onto sporangia formed on HAT agar or by suspending the sporangia harvested from the agar surface in 25 mM histidine solution and incubating the suspension for 1 h. Under the latter conditions, sporangia appear phase-bright immediately after suspension in histidine solution, and then the sporangium membrane gradually becomes transparent before spore release when observed by phase-contrast microscopy. Contrary to the dehiscence process of the wild-type sporangia, sporangium dehiscence was severely repressed, and zoospores were scarcely observed in the Δ*asfR* strain (Fig. 3; Fig. S4). Hence, we quantified the spores released from sporangia (under the former conditions) in the wild-type and Δ*asfR* strains, both of which contained the empty vector pCAM2 (22), by counting the colonies formed on YBNM agar after incubation at 30°C for 2 days. In this experiment, all colonies formed on YBNM agar came from the spores released from sporangia because the solution containing the zoospores was filtered through a 5 µm membrane filter to eliminate hyphae and sporangia. The wild-type sporangia formed on a single HAT agar plate released $10^7$

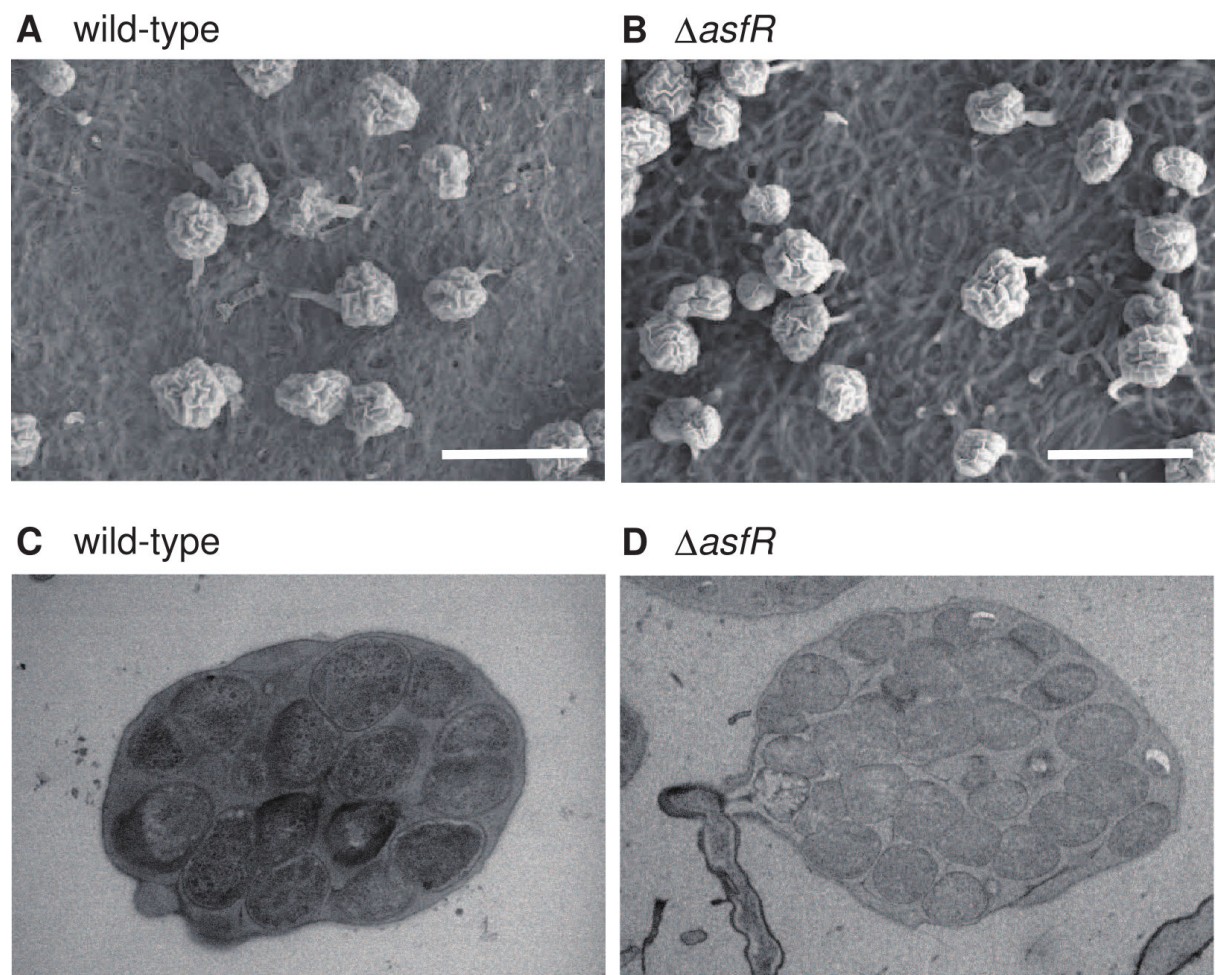

**A** wild-type
**B** Δ*asfR*
**C** wild-type
**D** Δ*asfR*

**FIG 2** Electron microscopic observation of sporangia. (A, B) SEM observation of sporangia produced on HAT agar after 7 days of cultivation. (A) Wild-type strain. (B) Δ*asfR* strain. Scale bars, 10 μm. (C, D) TEM observation of ultrathin sections of sporangia produced on HAT agar after 7 days of cultivation. (C) Wild-type strain. (D) Δ*asfR* strain. Scale bars, 1 μm.

spores (Fig. 4). In contrast, the Δ*asfR* sporangia released only $10^3$ spores per plate (Fig. 4), which is consistent with the phase-contrast microscopy observations (Fig. 3). In a gene complementation test, the introduction of pCAM2 carrying *asfR* into the Δ*asfR* strain resulted in almost complete restoration of the number of spores released from the sporangia (Fig. 4). These results clearly indicate that AsfR is crucial for the formation of physiologically mature sporangia that can release spores under sporangium dehiscence-inducing conditions.

## A putative phosphorylation site is essential for AsfR function

Previous studies have reported that several atypical receiver domain proteins function in a phosphorylation-independent manner, irrespective of the presence of the Asp phosphorylation site in their amino acid sequences (34). These proteins have been grouped into the PIARR (Phosphorylation-Independent Activation of Response Regulator) class (34). For instance, NblR, a member of the PIARR class, has been implicated in the general adaptation to stress in *Synechococcus* sp. PCC7942, and an NblR mutant, in which the Asp phosphorylation site is replaced by a nonphosphorylatable Ala, remained functional (34).

As described above, Asp-72 is the predicted phosphorylation site of AsfR. To validate the importance of this phosphorylation site for the physiological function of AsfR, we

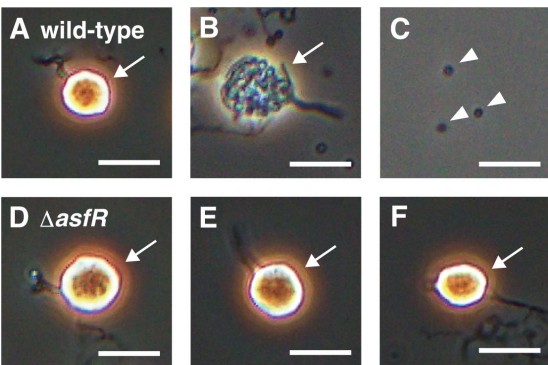

**FIG 3** Observation of sporangium dehiscence using phase-contrast microscopy. Sporangia produced on HAT agar were harvested and suspended in 25 mM histidine solution to induce sporangium dehiscence. Micrographs of the wild-type (A–C) and Δ*asfR* (D–F) strains are shown. Images in (A) and (D) were obtained immediately after the suspension. Images in (B) and (E) were obtained 15 min after the suspension. Images in (C) and (F) were obtained 30 min after the suspension. In the wild-type strain, sporangia appeared phase-bright immediately after the suspension (A) and the sporangium membrane gradually became transparent before spore release (B). Sporangia (including those whose membrane became transparent) and released spores are indicated by arrows and arrowheads, respectively. Scale bars, 10 µm. The entire images of each microscopic field are shown in Fig. S4.

generated and introduced a mutated *asfR* gene encoding AsfR (D72N) into the Δ*asfR* strain. As expected, the number of spores released from the sporangia was not restored by the introduction of this mutated gene (Fig. 4). This result clearly indicates that the D72N mutation renders AsfR non-functional with respect to the phenotypic changes observed in this study.

## Search for the partner HK of AsfR

As described above, we considered AsfR an orphan RR. A total of 88 putative HK and RR gene pairs were annotated in the *A. missouriensis* genome sequence, and 50 genes were annotated to encode putative orphan HKs (Table S2). Among these, 17 gene products seem to be putative hybrid HKs because they possess RR receiver domains (Table S2). Therefore, we assumed that one or more of the remaining 33 putative HKs constitute

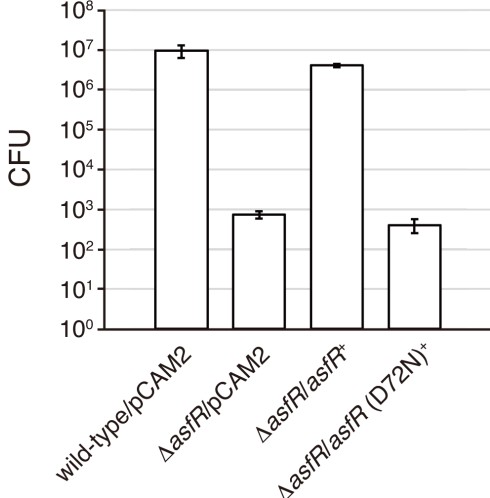

**FIG 4** Number of spores released from sporangia. Zoospores released from sporangia by pouring 25 mM $NH_4HCO_3$ solution were counted as CFU on YBNM agar. Data are presented as the mean values ± standard errors of three biological replicates.

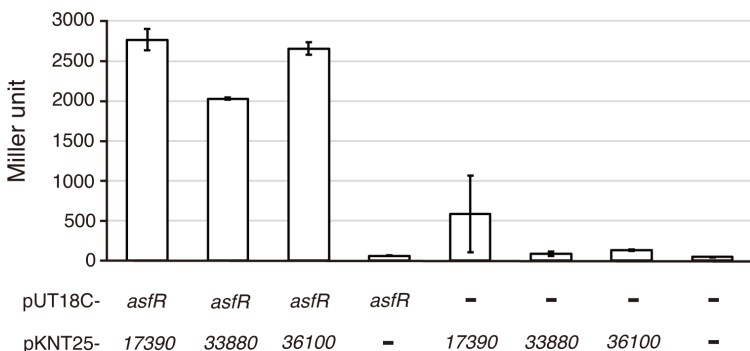

**FIG 5** Bacterial two-hybrid assay of AsfR and three HKs. β-Galactosidase activities (Miller units) of *E. coli* BTH101 co-transformed with plasmids producing AsfR and each of AMIS_17390, AMIS_33880, and AMIS_36100 are shown. Empty vectors expressing the T18 or T25 domains of adenylate cyclase were used as vector controls. The values represent mean values ± standard errors from four biological replicates.

a functional TCS with AsfR. To test this possibility, we performed bacterial adenylate cyclase-based two-hybrid (BACTH) assays using *E. coli* as a host because the BACTH assay has been utilized to detect the interactions between HKs and RRs (35, 36). In the BACTH assay using agar plates, we observed weak interactions between AsfR and each of the three putative HKs, AMIS_17390, AMIS_33880, and AMIS_36100, which were indicated by the slow growth of blue colonies of the transformants (Fig. S5). Thus, we quantified protein-protein interactions via a β-galactosidase assay using the same transformants in liquid culture. As expected, we detected significant increases in β-galactosidase activity in *E. coli* cells producing both AsfR and each of the three HKs, compared to those harboring empty vectors (Fig. 5).

Considering that the AsfR (D72N) mutant protein was non-functional, a knockout mutant of the gene encoding an orphan HK that phosphorylates AsfR is expected to show phenotypic changes similar to those of the Δ*asfR* strain. Thus, we constructed single-gene, double-gene, and triple-gene deletion mutant strains of these three HK genes in all possible combinations and analyzed their phenotypes. Contrary to our expectations, all mutant strains showed no phenotypic changes in sporangium formation, sporangium dehiscence, or release of spores from sporangia compared to the wild-type strain (Fig. S6 to S8). These results indicate that these HKs, at least solely by themselves, are not responsible for the phosphorylation of AsfR in *A. missouriensis*.

## DISCUSSION

In this study, we identified AsfR, an orphan RR receiver domain protein of the TCS, as a key regulator of the formation of physiologically mature sporangia that can release spores under sporangium dehiscence-inducing conditions. As described in the Introduction, stand-alone receiver domain proteins play their roles by binding to other proteins or by carrying a phosphoryl group between sensor HKs and Hpt proteins in multistep phosphorelays. In the *A. missouriensis* genome, *AMIS_37460* encodes a single-domain protein composed of the Hpt domain, and five genes (*AMIS_18620*, *AMIS_25430*, *AMIS_36070*, *AMIS_68710*, and *AMIS_76570*) encode multidomain proteins with the Hpt domain. Previously, we demonstrated that none of these six genes is required for sporangium formation and dehiscence by gene disruption (13). If AsfR exerts its influence by mediating phosphoryl transfer to an Hpt protein, a null mutant strain of the Hpt protein-encoding gene is expected to show phenotypic changes similar to those observed in the Δ*asfR* strain. Therefore, we assume that AsfR binds to other protein(s) to modulate the functions of the target protein(s), although we cannot exclude the possibility that two or more Hpt domain-containing proteins mediate the phosphotransfer reaction between AsfR and its partner RR(s).

BACTH assays indicated that three orphan HKs, AMIS_17390, AMIS_33880, and AMIS_36100, interact with AsfR. However, gene disruption experiments revealed that these putative HKs were not essential for sporangium dehiscence in *A. missouriensis*. Because Asp-72, the predicted phosphorylation site of AsfR, was shown to be essential for AsfR function, we assumed that there is an HK(s) that phosphorylates AsfR. We speculate that our screening for a kinase that phosphorylates AsfR using the BACTH assay missed the true target.

Although *asfR* transcription was activated during sporangium formation (Fig. S3), the Δ*asfR* strain produced normally shaped sporangia that contain apparently normal dormant spores in similar sizes (Fig. 2). Therefore, we assume that *A. missouriensis* activates genes required for sporangium formation and genes involved in sporangium dehiscence in parallel during sporangium formation. This refined gene regulatory program appears to enable the prompt release of spores from sporangia under sporangium dehiscence-inducing conditions. Considering the severe defect in sporangium dehiscence in the Δ*asfR* strain, AsfR and the target protein(s) of AsfR seem to play a key role in the formation of physiologically mature sporangia that are fully prepared to release spores under sporangium dehiscence-inducing conditions; in other words, they play a key role in the preparation of the onset and progression of sporangium dehiscence. Inevitably, the identification of the target protein(s) of AsfR is an important subject for future research. Notably, a gene knockout mutant of *bldC*, which encodes a global transcriptional activator, showed a sporangium dehiscence-deficient phenotype similar to that of the Δ*asfR* strain (37). Many genes seem to be involved in the formation of physiologically mature sporangia that are fully prepared to release spores, and there seems to be a complex regulatory system in which AsfR and BldC play key regulatory roles.

## ACKNOWLEDGMENTS

We would like to thank Tatsuki Koyama for his help in plasmid construction for bacterial two-hybrid assays. This research was supported in part by Grants-in-Aid for Scientific Research (A) (JP26252010), (B) (JP18H02122), and (C) (JP17K07711 and JP20K05781), and a Grant-in-Aid for Scientific Research on Innovative Areas (JP19H05685) from the Ministry of Education, Culture, Sports, Science, and Technology of Japan.

This work was also supported in part by the Japan Society for the Promotion of Science (JSPS) A3 Foresight Program.

## AUTHOR AFFILIATIONS

[1]Department of Biotechnology, Graduate School of Agricultural and Life Sciences, The University of Tokyo, Bunkyo-ku, Tokyo, Japan
[2]Bioimaging Center, Graduate School of Frontier Sciences, The University of Tokyo, Kashiwa-shi, Chiba, Japan
[3]Collaborative Research Institute for Innovative Microbiology, The University of Tokyo, Bunkyo-ku, Tokyo, Japan

## AUTHOR ORCIDs

Zhuwen Tan  http://orcid.org/0009-0002-8674-0263
Takeaki Tezuka  http://orcid.org/0000-0002-9042-2674
Yasuo Ohnishi  http://orcid.org/0000-0001-7633-9236

## FUNDING

| Funder | Grant(s) | Author(s) |
| --- | --- | --- |
| Ministry of Education, Culture, Sports, Science and Technology (MEXT) | JP26252010 | Yasuo Ohnishi |

| Funder | Grant(s) | Author(s) |
|---|---|---|
| Ministry of Education, Culture, Sports, Science and Technology (MEXT) | JP18H02122 | Yasuo Ohnishi |
| Ministry of Education, Culture, Sports, Science and Technology (MEXT) | JP17K07711 | Takeaki Tezuka |
| Ministry of Education, Culture, Sports, Science and Technology (MEXT) | JP20K05781 | Takeaki Tezuka |
| Ministry of Education, Culture, Sports, Science and Technology (MEXT) | JP19H05685 | Yasuo Ohnishi |
| MEXT \| Japan Society for the Promotion of Science (JSPS) | A3 Foresight Program | Yasuo Ohnishi |

## AUTHOR CONTRIBUTIONS

Takuya Akutsu, Investigation, Methodology | Zhuwen Tan, Investigation, Methodology | Aiko Hirata, Investigation, Methodology | Takeaki Tezuka, Funding acquisition, Investigation, Writing – original draft, Writing – review and editing | Yasuo Ohnishi, Funding acquisition, Project administration, Writing – review and editing

## ADDITIONAL FILES

The following material is available online.

### Supplemental Material

**Supplemental figures and tables (Spectrum03272-24-s0001.pdf).** Figures S1 to S8; Tables S1 and S2.

### Open Peer Review

**PEER REVIEW HISTORY (review-history.pdf).** An accounting of the reviewer comments and feedback.

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
