## [Reviewer comments · Microbiology Spectrum]

Microbiology Spectrum

Involvement of an orphan response regulator of the two-component regulatory system in the formation of physiologically mature sporangia in *Actinoplanes missouriensis*

Takuya Akutsu, Zhuwen Tan, Aiko Hirata, Takeaki Tezuka, and Yasuo Ohnishi

Corresponding Author(s): Takeaki Tezuka, Tokyo Daigaku

Review Timeline:

Submission Date:	December 13, 2024
Editorial Decision:	January 14, 2025
Revision Received:	January 18, 2025
Accepted:	February 7, 2025

Editor: Beile Gao

Reviewer(s): The reviewers have opted to remain anonymous.

Transaction Report:

DOI: <https://doi.org/10.1128/spectrum.03272-24>

Re: Spectrum03272-24 (Involvement of an orphan response regulator of the two-component regulatory system in the formation of physiologically mature sporangia in *Actinoplanes missouriensis*)

Dear Dr. Takeaki Tezuka:

Thank you for the privilege of reviewing your work. Below you will find my comments, instructions from the Spectrum editorial office, and the reviewer comments.

Revision Guidelines

Sincerely,
Beile Gao
Editor
Microbiology Spectrum

Reviewer #1 (Comments for the Author):

The authors reported the identification of an orphan response regulator of two-component system involved in the regulation of the formation of physiologically mature sporangia in the bacterium, *Actinoplanes missouriensis* using genetic and morphological approaches. This study is well-done and the data is convincing. This manuscript is well written and easy to understand. I recommend its publication in this journal. I have only two minor comments.

1) I recommend the authors to use the following Abbreviations throughout the manuscript, such as RR for response regulator, TCS for two-component regulatory system, HK for histidine kinase, etc.

2) There are fewer images in the articles. So, I recommend to move Figure S1-S4 from the supplementary material to the manuscript and combine together in Figure 1.

Reviewer #2 (Comments for the Author):

This is a well-written and short report that identifies a previously unknown gene that is required for dehiscence of sporangia of the actinomycete *Actinoplanes missouriensis*, i.e. the water-induced release of motile zoospores from the unique type of sporangia formed by this organism. The gene, *asfR*, encodes a single-domain response regulator, and the authors show by mutagenesis that the Asp residue that normally is phosphorylated in response regulators is required for function of the protein, suggesting that phosphorylation of *AsfR* is important for sporangium dehiscence. They screen for interacting histidine kinases by testing protein-protein interaction in a two-hybrid system, but none of the histidine kinases that give signal indicating interaction with *AsfR* in the BACTH assays are needed for dehiscence. Thus, it remains unknown how *AsfR* is phosphorylated. In addition, the paper includes structural modelling of *AsfR* using AlphaFold, survey of *asfR* genes across the *Actinoplanes* genus, and illustration of transcription of *asfR* through the developmental cycle, based on a previously published RNA-seq data set. Overall, it is a rather thin paper that essentially identifies *AsfR* and demonstrates that it (and likely its phosphorylation) is required specifically for sporangium dehiscence, but not for other parts of the life cycle, like formation of spores or sporangia. The search for the cognate histidine kinase is not successful and therefore of limited value. However, the discovery of *AsfR* and its role in this unusual and original developmental system is of value for the field and will hopefully spur further and more mechanistic investigations.

Except for the limited scope of the paper, what is presented is clearly presented and of good quality. I have only a few relatively minor comments.

1. Have *AsfR* homologues been found in any genera outside *Actinoplanes*? The paper only discusses *Actinoplanes*.
2. The Introduction could include some general references or reviews regarding both relevant aspects of two-component systems and the *Actinoplanes* developmental biology.
3. It would help the reader if it was briefly mentioned what the basis for counterselection is for the vector pK19mobApr (lines 138-144).
4. How was the D72N mutation introduced by PCR? (line 163)
5. The basis for shading (grey and black) in Fig. S4 is not clear and should be mentioned in the legend.

Point-by-point response to reviewers' comments

Reviewer #1 (Comments for the Author):

The authors reported the identification of an orphan response regulator of two-component system involved in the regulation of the formation of physiologically mature sporangia in the bacterium, *Actinoplanes missouriensis* using genetic and morphological approaches. This study is well-done and the data is convincing. This manuscript is well written and easy to understand. I recommend its publication in this journal. I have only two minor comments.

*Thank you very much for your positive comment.

1) I recommend the authors to use the following abbreviations throughout the manuscript, such as RR for response regulator, TCS for two-component regulatory system, HK for histidine kinase, etc.

*According to this comment, we have used three abbreviations, HK, RR, and TCS for histidine kinase, response regulator, and two-component system, respectively, throughout the revised manuscript.

2) There is fewer images in the articles. So, I recommend to move Figure S1-S4 from the supplementary material to the manuscript and combine together in Figure 1.

*Thank you very much for this constructive suggestion. We carefully considered on this point and decided to move Figs. S2 and S3 to the main manuscript as a new Figure 1, but keep Figs. S1 and S4 in the supplemental material.

Reviewer #2 (Comments for the Author):

This is a well-written and short report that identifies a previously unknown gene that is required for dehiscence of sporangia of the actinomycete *Actinoplanes missouriensis*, i.e. the water-induced release of motile zoospores from the unique type of sporangia formed by this organism. The gene, *asfR*, encodes a single-domain response regulator, and the authors show by mutagenesis that the Asp residue that normally is phosphorylated in response regulators is required for function of the protein, suggesting that phosphorylation of AsfR is important for sporangium dehiscence. They screen for interacting histidine kinases by testing protein-protein interaction in a two-hybrid system, but none of the histidine kinase that give signal indicating interaction with AsfR in the BACTH assays are needed for dehiscence. Thus, it remains unknown how AsfR is phosphorylated. In addition, the paper includes structural modelling of

AsfR using AlphaFold, survey of *asfR* genes across the *Actinoplanes* genus, and illustration of transcription *asfR* through the developmental cycle, based on a previously published RNA-seq data set. Overall, it is a rather thin paper that essentially identifies AsfR and demonstrates that it (and likely its phosphorylation) is required specifically for sporangium dehiscence, but not for other parts of the life cycle, like formation of spores or sporangia. The search for the cognate histidine kinase is not successful and therefore of limited value. However, the discovery of AsfR and its role in this unusual and original developmental system is of value for the field and will hopefully spur further and more mechanistic investigations.

Except for the limited scope of the paper, what is presented is clearly presented and of good quality. I have only a few relatively minor comments.

*Thank you very much for your careful reading and positive comment.

1. Have AsfR homologues been found in any genera outside *Actinoplanes*? The paper only discusses *Actinoplanes*.

*According to this comment, we searched for the AsfR homologs in species of other genera and found that AsfR homologs are conserved in related actinomycetes including members of the genera *Patulibacter*, *Pseudosporangium*, *Couchioplanes*, *Nucisporomicrobium*, *Spirilliplanes*, *Krasilnikovia*, *Paractinoplanes*, *Symbioplanes*, and *Winogradskya* (more than 53% amino acid sequence identity). Meanwhile, we found that the AsfR homologs are not conserved in members of the genus *Streptomyces* (less than 35% identity), which have been most extensively investigated about morphological development in actinomycetes. We have added a description of this result in the revised manuscript as follows (L. 235–252; Fig. S2B).

“Furthermore, proteins showing high similarities to AsfR are also conserved in other actinomycete species including *Patulibacter* sp. NPD 049589, *Pseudosporangium ferrugineum*, *Couchioplanes caeruleus*, *Nucisporomicrobium flavum*, *Spirilliplanes yamanashiensis*, *Krasilnikovia cinnamomea*, *Paractinoplanes lichenicola*, *Symbioplanes lichenis*, *Winogradskya consettensis* (Fig. S2B). Members of the genus *Patulibacter* form motile cells with long flagella (26). *C. caeruleus* forms flagellated arthrospores by fragmentation of aerial hyphae (27). *N. flavum* forms irregular pseudosporangia consisting of non-motile spores (28). *S. yamanashiensis* forms short chains of spores from aerial hyphae arranged in spirals (29). *K. cinnamomea* forms pseudosporangia on short sporangiophores above the surface of substrate mycelium (30). Members of the genera *Paractinoplanes*, *Symbioplanes*, and *Winogradskya* are closely related to *Actinoplanes* (31). These homologs share at least 53% amino acid sequence identity with AsfR, demonstrating that AsfR homologs are evolutionarily conserved among the genus *Actinoplanes* and related actinomycetes (Fig. S2). Meanwhile, AsfR homologs are not conserved in members of the genus *Streptomyces* (less than 35% amino acid sequence identity), which forms branched

substrate mycelia during vegetative growth and subsequently produces aerial hyphae that culminate in non-motile exospores (32).”

2. The Introduction could include some general references or reviews regarding both relevant aspects of two-component systems and the *Actinoplanes* developmental biology.

*According to this comment, we have added eight references concerning the two-component system and life cycle of *A. missouriensis* in the Introduction of the revised manuscript (references 1, 2, 4–7, 9, and 10).

3. It would help the reader if it was briefly mentioned what the basis for counterselection is for the vector pK19mobApr (lines 138-144).

*According to this comment, we have added a brief description of the counterselection in the revised manuscript as follows (L. 144–146).

“*A. missouriensis* cells expressing *sacB*, encoding levansucrase, from pK19mobApr exhibit lethal sensitivity to sucrose.”

4. How was the D72N mutation was introduced by PCR? (line 163)

*We have added a description of the site-directed mutagenesis in the Materials and Methods of the revised manuscript as follows (L. 163–168).

“To construct the mutated *asfR* gene, pCAM2 carrying *asfR* was used for site-directed mutagenesis. The D72N mutated DNA fragment was amplified by overlap extension PCR using the primer pairs 76070_comp-F and 76070_D72N-R, and 76070_D72N-F and 76070_comp-R (Table S1). The amplified DNA fragment was digested with EcoRI and XbaI, and cloned into pUC19 digested with the same restriction enzymes.”

5. The basis for shading (grey and black) in Fig. S4 is not clear and should be mentioned in the legend.

*Thank you very much for your careful reading. For simplicity, we have revised the alignment by showing the identical amino acid residues in black background and the remaining residues without background color in the revised manuscript (Figs. 1 and S2). We have added the description in the figure legends.

Re: Spectrum03272-24R1 (Involvement of an orphan response regulator of the two-component regulatory system in the formation of physiologically mature sporangia in *Actinoplanes missouriensis*)

Dear Dr. Takeaki Tezuka:

Your manuscript has been accepted, and I am forwarding it to the ASM production staff for publication. Your paper will first be checked to make sure all elements meet the technical requirements. ASM staff will contact you if anything needs to be revised before copyediting and production can begin. Otherwise, you will be notified when your proofs are ready to be viewed.

Sincerely,
Beile Gao
Editor
Microbiology Spectrum

Reviewer #1 (Comments for the Author):

The authors have addressed all of my comments raised previously.